# GRAPH-LEVEL REPRESENTATION LEARNING WITH JOINT-EMBEDDING PREDICTIVE ARCHITECTURES

## ABSTRACT

Joint-Embedding Predictive Architectures (JEPAs) have recently emerged as a novel and powerful technique for self-supervised representation learning. They aim to learn an energy-based model by predicting the latent representation of a target signal $y$ from a context signal $x$. JEPAs bypass the need for data augmentation and negative samples, which are typically required by contrastive learning, while avoiding the overfitting issues associated with generative-based pretraining. In this paper, we show that graph-level representations can be effectively modeled using this paradigm and propose Graph-JEPA, the first JEPA for the graph domain. In particular, we employ masked modeling to learn embeddings for different subgraphs of the input graph. To endow the representations with the implicit hierarchy that is often present in graph-level concepts, we devise an alternative training objective that consists of predicting the coordinates of the encoded subgraphs on the unit hyperbola in the 2D plane. Extensive validation shows that Graph-JEPA can learn representations that are expressive and competitive in both graph classification and regression problems. The implementation will be available upon acceptance.

## 1 INTRODUCTION

Graph data is ubiquitous in the real world due to its ability to universally abstract various concepts and entities (Ma & Tang, 2021; Veličković, 2023). To deal with this peculiar data structure, Graph Neural Networks (GNNs) (Scarselli et al., 2008; Kipf & Welling, 2016a; Gilmer et al., 2017; Veličković et al., 2017) have established themselves as a staple solution. Nevertheless, most applications of GNNs usually rely on ground-truth labels for training. The growing amount of graph data in fields such as bioinformatics, chemoinformatics, and social networks has made manual labeling laborious, sparking significant interest in unsupervised graph representation learning.

A particularly emergent area in this line of research is self-supervised learning (SSL). In SSL, alternative forms of supervision are created stemming from the input signal. This process is then typically followed by invariance-based or generative-based pretraining (Liu et al., 2023; Assran et al., 2023). Invariance-based approaches optimize the model to produce comparable embeddings for different views of the input signal. A common term associated with this procedure is contrastive learning (Tian et al., 2020). Typically, these alternative views are created by a data augmentation procedure. The views are then passed through their respective encoder networks (which may share weights), as shown in Figure 1a. Finally, an energy function, usually framed as a distance, acts on the latent embeddings. In the graph domain, several works have applied contrastive learning by designing graph-specific augmentations (You et al., 2020), using multi-view learning (Hassani & Khasahmadi, 2020) and even adversarial learning (Suresh et al., 2021). Invariance-based pretraining is effective but comes with several drawbacks i.e., the necessity to augment the data and process negative samples, which limits computational efficiency. In order to learn embeddings that are useful for downstream tasks, the augmentations must also be non-trivial.

Generative-based pretraining methods on the other hand typically remove or corrupt portions of the input and predict them using an autoencoding procedure (Vincent et al., 2010; He et al., 2022), or rely on autoregressive modeling (Brown et al., 2020; Hu et al., 2020). Figure 1b depicts the typical instantiation of these methods: The input signal $x$ is fed into an encoder network that constructs the latent representation and a decoder generates $\hat{y}$, the data corresponding to the target signal $y$. The

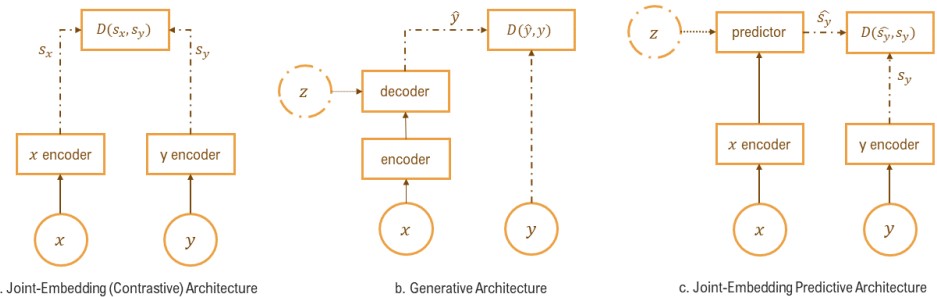

Figure 1: Illustration of the three main SSL approaches: (a) Joint-Embedding Architectures learn to create similar embeddings for inputs x and y that are compatible with each other and dissimilar embeddings for inputs that are not compatible. This compatibility is implemented in practice by creating different views of the input data. (b) Generative Architectures reconstruct a signal $y$ from a compatible signal $x$ by conditioning the decoder network on additional (potentially latent) variables $z$. (c) Joint-Embedding Predictive Architectures act as a bridge: They utilize a predictor network that processes the context $x$ and is conditioned on additional (potentially latent) factors, to predict the embedding of the target $y$ in latent space.

energy function is then applied in data space, often as a reconstruction term (Bengio et al., 2013). Generative models generally display strong overfitting tendencies since they have to estimate the data distribution (implicitly or explicitly), so the latent representations must be directly descriptive of the whole data space. This can be very problematic for graphs given that they live in a non-Euclidean and inhomogenous data space. Nevertheless, masked autoencoding has recently shown promising results in the graph domain with appropriately designed models (Hou et al., 2022; Tan et al., 2023).

Inspired by the innovative Joint-Embedding Predictive Architecture (JEPA) (LeCun, 2022; Assran et al., 2023), we propose Graph-JEPA, the first JEPA for the graph domain, to learn graph-level representations by bridging contrastive and generative models. As illustrated in Figure 1c, a JEPA has two encoder networks that receive the input signals and produce the corresponding representations. Notably, the two encoders can be different models and don't need to share weights. A predictor module outputs a prediction of the latent representation of one signal based on the other, possibly conditioned on another variable. Graph-JEPA does not require any negative samples or data augmentation and by operating in the latent space it avoids the pitfalls associated with learning high-level details needed to fit the data distribution. However, the graph domain presents us with additional challenges, namely: context and target extraction; designing a latent prediction task that is optimal for graph-level concepts; and learning expressive representations. In response to these questions, we equip Graph-JEPA with a specific masked modeling objective. The input graph is first divided into several subgraphs, and then the latent representation of randomly chosen target subgraphs is predicted given a context subgraph. The subgraph representations are consequently pooled to create a graph-level representation that can be used for downstream tasks.

The nature of graph-level concepts is often assumed to be hierarchical (Ying et al., 2018), thus we conjecture that the typical latent reconstruction objective used in current JEPA formulations is not enough to provide optimal downstream performance. To this end, we design a prediction objective that starts by expressing the target subgraph encoding as a high-dimensional description of the hyperbolic angle. The predictor module is then tasked with predicting the location of the target in the 2D unit hyperbola. This prediction is compared with the target coordinates, obtained by using the aforementioned hyperbolic angle. Graph-JEPA outperforms popular contrastive and generative graph-level SSL methods on different datasets, while maintaining efficiency and ease of training. Notably, we observe from our experiments that Graph-JEPA can run up to 1.45x faster than Graph-MAE (Hou et al., 2022) and 8x faster than MVGRL (Hassani & Khasahmadi, 2020). Finally, we empirically demonstrate Graph-JEPA's ability to learn highly expressive graph representations, showing it almost perfectly distinguishes pairs of non-isomorphic graphs that the 1-WL test cannot differentiate.

## 2 RELATED WORK

### 2.1 SELF-SUPERVISED GRAPH REPRESENTATION LEARNING

Graph Neural Networks (GNNs) (Scarselli et al., 2008; Kipf & Welling, 2016a; Veličković et al., 2017; Hamilton et al., 2017; Xu et al., 2018; Wu et al., 2019) are now established solutions to different graph machine learning problems such as node classification, link prediction, and graph classification. Nevertheless, the cost of labeling graph data is quite high given the immense variability of graph types and the information they can represent. To alleviate this problem, SSL on graphs has become a research frontier. SSL methods on graphs can be divided into two major groups (Xie et al., 2022b; Liu et al., 2023):

**Contrastive Methods.** Contrastive learning methods usually consist of minimizing an energy function (Hinton, 2002; Gutmann & Hyvärinen, 2010) between different views of the same data. InfoGraph (Sun et al., 2019) maximizes the mutual information between the graph-level representation and the representations of substructures at different scales. GraphCL (You et al., 2020) works similarly to distance-based contrastive methods in the imaging domain. The authors first propose four types of graph augmentations and then perform contrastive learning based on them. The work of Hassani & Khasahmadi (2020) goes one step further by contrasting structural views of graphs. They also show that a large number of views or multiscale training does not seem to be beneficial, contrary to the image domain. Another popular research direction for contrastive methods is learning graph augmentations (Suresh et al., 2021) and AutoSSL (Jin et al., 2021), where the goal is to learn how to automatically leverage multiple pretext tasks effectively. Contrastive learning methods typically require a lot of memory due to data augmentation and negative samples. Graph-JEPA is much more efficient than typical formulations of these architectures given that it does not require any augmentations or negative samples. Another major difference is that the prediction in latent space in JEPAs is done through a separate predictor network, rather than using the common Siamese structure (Bromley et al., 1993)(Figure 1a vs c).

**Generative Methods.** The goal of generative models is to recover the data distribution, an objective that is typically implemented through a reconstruction process. In the graph domain, most generative architectures that are also used for SSL are extensions of Auto-Encoder (AE) (Hinton & Zemel, 1993) and Variational Auto-Encoder (VAE) (Kingma & Welling, 2013) architectures. These models learn an embedding from the input data and then use a reconstruction objective with (optional) regularization in order to maximize the data evidence. Kipf & Welling (2016b) extended the framework of AEs and VAEs to graphs by using a GNN as an encoder and the reconstruction of the adjacency matrix as a training objective. However, the results on node and graph classification benchmarks with these embeddings are often unsatisfactory compared with contrastive learning methods, a tendency also observed in other domains (Liu et al., 2023). A recent and promising direction is masked autoencoding (MAE) (He et al., 2022), which has proved to be a very successful framework for the image and text domains. GraphMAE (Hou et al., 2022) is an instantiation of MAEs in the graph domain, where the node attributes are perturbed and then reconstructed, providing a paradigm shift from the structure learning objective of GAEs. S2GAE (Tan et al., 2023) is one of the latest GAEs, which focuses on reconstructing the topological structure but adds several auxiliary objectives and additional designs. Our architecture differs from generative models in that it learns to predict directly in the latent space, thereby bypassing the necessity of remembering and overfitting high-level details that help maximize the data evidence (Figure 1b vs c).

### 2.2 JOINT-EMBEDDING PREDICTIVE ARCHITECTURES

Joint-Embedding Predictive Architectures (LeCun, 2022) are a recently proposed design for SSL. The idea is similar to both generative and contrastive approaches, yet JEPAs are non-generative since they cannot directly predict $y$ from $x$, as shown in Figure 1c. The energy of a JEPA is given by the prediction error in the embedding space, not the input space. These models can therefore be understood as a way to capture abstract dependencies between $x$ and $y$, potentially given another latent variable $z$. It is worth noting that the different models comprising the architecture may differ in terms of structure and parameters. An in-depth explanation of Joint-Embedding Predictive Architectures and their connections to human representation learning is provided by LeCun (2022). The most well-known works to use particular instantiations of JEPAs before the term was officially coined are BYOL (Grill et al., 2020) and SimSiam (Chen & He, 2021). Inspired by these trends, there has been

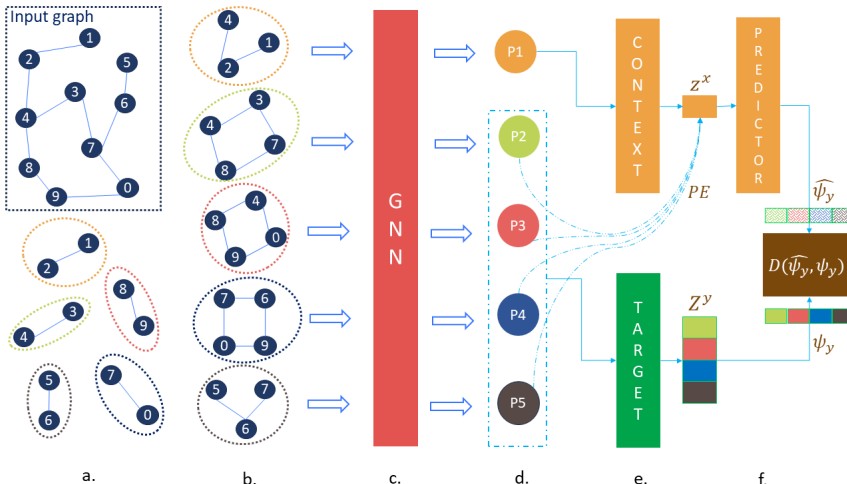

Figure 2: An overview of Graph-JEPA. We first extract non-overlapping subgraphs (patches) (a.), perform a 1-hop neighborhood expansion (b.), and encode the subgraphs with a GNN (c.). After the subgraph encoding, one is randomly picked as the context and $m$ others as the targets (d.) and they are fed into their respective encoders (e.). The embeddings generated from the target encoder are used to produce the target subgraphs' coordinates $\psi_y$. Finally, the predictor network is tasked with directly predicting the coordinates $\hat{\psi}_y$ for each target subgraph based on the context embedding and the positional embedding of each target subgraph (f.). A regression loss acts as the energy function $D$ between the predicted and target coordinates.

recent work that tries to employ latent prediction objectives for graph SSL, showing advantages compared to contrastive learning. Zhang et al. (2021) rely on ideas from Canonical Correlation Analysis to frame a learning objective that preserves feature invariance and forces decorrelation when necessary, while Xie et al. (2022a) performs learns by doing both instance-level reconstruction and feature-level invariance. Differently from these two methods, Graph-JEPA operates only in the latent space with a single energy function frame as a reconstruction of the learned features. There have recently been two works that implement JEPAs in practice, both in the vision domain: I-JEPA (Assran et al., 2023) and MC-JEPA (Bardes et al., 2023). I-JEPA provides an instantiation in the context of images, where the goal is to predict the latent embeddings of different image patches given a context patch. MC-JEPA instantiates the architecture in the context of learning optical flow and content features within a shared encoder. We propose the first JEPA for the graph domain and use it to learn graph-level representations in a self-supervised manner.

## 3 METHOD

**Preliminary.** Let a graph $G$ be defined as $G = (V, E)$ where $V = \{v_1 \ldots v_N\}$ is the set of nodes, with a cardinality $|V| = N$, and $E = \{e_1 \ldots e_M\}$ is the set of edges, with a cardinality $|E| = M$. We consider symmetric, unweighted graphs, although our method can be generalized to directed or weighted graphs. In this setting, $G$ can be represented by an adjacency matrix $A \in \{0, 1\}^{N \times N}$, with $A_{ij} = 1$ if nodes $v_i$ and $v_j$ are connected and $A_{ij} = 0$ otherwise.

**A general overview.** Figure 2 provides an overview of the proposed architecture. The high-level idea of Graph-JEPA is to first divide the input graph into subgraphs (patches) (He et al., 2023) and then predict the representation of a randomly chosen target subgraph from a context subgraph (Assran et al., 2023). We would like to stress that this masked modeling objective is realized in latent space, without the need for augmentations or negative samples. The subgraph representations are then pooled to create a vectorized graph-level representation. Therefore Graph-JEPA can be described through a sequence of operations, namely: 1. Spatial Partitioning; 2. Subgraph Embedding; 3. Context and Target Encoding; 4. Latent Target Prediction.

## 3.1 SPATIAL PARTITIONING

We base the initial part of the Graph-JEPA architecture on the recent work of He et al. (2023), but similar ideas consisting of graph partitioning have been proposed before for Graph SSL (Jin et al., 2020). This step consists of creating different subgraphs (patches) of a graph, similar to how Vision Transformers (ViT) (Dosovitskiy et al., 2020) operate on images. We rely on the METIS (Karypis & Kumar, 1998) graph clustering algorithm, which partitions a graph into a pre-defined, non-overlapping number of clusters $p$ (Figure 2a), such that the number of within-cluster links is much higher than between-cluster links. Note that having non-overlapping subgraphs can be problematic since edges can be lost in this procedure and it is possible to end up with empty "subgraphs". We would also like to preserve the notion of locality in each subgraph while relating it to others that are close in terms of shared nodes. To create this overlap and avoid completely empty subgraphs, a one-hop neighborhood expansion of all the nodes belonging to a subgraph is performed (Figure 2b).

## 3.2 SUBGRAPH EMBEDDING

After partitioning the graph, we learn a representation for each subgraph through a GNN (Figure 2c.). The specific choice of GNN architecture is arbitrary and depends on what properties we wish to induce in the representation. The learned node embeddings are mean pooled to create a vector representation for each subgraph: $\{h_1...h_p\}, h \in \mathbb{R}^d$. Given that these embeddings will be used as context or target variables, it is important to provide additional information regarding the subgraphs to help guide the predictor network. Thus, we propose to use a positional embedding for each subgraph, which is implemented as the maximum Random Walk Structural Embedding (RWSE) of all the nodes in that subgraph. In this way, the position is characterized in a global and consistent manner for each patch. Formally, a RWSE (Dwivedi et al., 2021) for a node $v$ can be defined as:

$$P_v = (M_{ii}, M_{ii}^2, \ldots, M_{ii}^k), P_v \in \mathbb{R}^k, \quad M^k = (D^{-1}A)^k \tag{1}$$

where $M^k$ is the random-walk transition matrix of order $k$ and $i$ is the index of node $v$ in the adjacency matrix. Therefore, $M_{ii}^k$ encodes the probability of node $v$ landing to itself after a $k$-step random walk. Given a subgraph $l$, let $V_l$ denote the set of nodes in $l$. The subgraph RWSE is then defined as:

$$P_l = \max_{v \in V_l} P_v \tag{2}$$

## 3.3 CONTEXT AND TARGET ENCODING

Given the subgraph representations and their respective positional embeddings, we frame the Graph-JEPA prediction task in a similar manner to I-JEPA (Assran et al., 2023). The goal of the network is to predict the latent embeddings of randomly chosen target subgraphs, given one random context subgraph. The prediction is conditioned on positional information regarding each subgraph. At each training step, we choose one random subgraph as the context $x$ and $m$ others as targets $Y = \{y_1, \ldots, y_m\}$ (Figure 2d). These subgraphs are processed by the context and target encoders (Figure 2e) which are parametrized by Transformer encoder blocks (without self-attention for the context) where normalization is applied at first (Xiong et al., 2020). The target encoder uses Hadamard self-attention (He et al., 2023), but other choices, such as the standard self-attention mechanism (Vaswani et al., 2017) are perfectly viable. We can summarize this step as:

$$z^x = E_{context}(x), z^x \in \mathbb{R}^d, \quad Z^y = E_{target}(Y), Z^y \in \mathbb{R}^{m \times d} \tag{3}$$

At this stage, we could use the predictor network to directly predict $Z^y$ from $z^x$. This is the typical formulation of JEPAs, also followed by Assran et al. (2023). We argue that learning how to organize concepts for abstract objects such as graphs or networks directly in Euclidean space is suboptimal. In the following section, we propose a simple trick to bypass this problem using the encoding and prediction mechanisms in Graph-JEPA. The following subsections and the ablations studies in Section 4.3 will provide additional insights.

## 3.4 LATENT TARGET PREDICTION

Different works in Neuroscience have shown the importance of learning hierarchically consistent concepts (Deco et al., 2021), especially during infancy and young age (Rosenberg & Feigenson,

2013). Networks in the real world often conform to some concept of hierarchy (Moutsinas et al., 2021) and this assumption is frequently used when learning graph-level representations (Ying et al., 2018). Thus, we conjecture that Graph-JEPA should operate in a hyperbolic space, where learned embeddings implicitly organize hierarchical concepts(Nickel & Kiela, 2017; Zhao et al., 2023). This gives rise to another issue: Hyperbolic (Poincaré) embeddings are known to have several tradeoffs related to dimensionality (Sala et al., 2018; Guo et al., 2022), which severely limits the expressive ability of the model. Given that graphs can describe very abstract concepts, high expressivity in terms of model parameters is preferred. In summary, we would ideally like to have a high-dimensional latent code that has a concept of hyperbolicity built into it.

To achieve this, we think of the target embedding as a high-dimensional representation of the hyperbolic angle, which allows us to describe each target patch through its position in the 2D unit hyperbola. Formally, given a target patch $l$, its embedding $Z_l^y$ and positional encoding $P_l$, we first express the latent target as:

$$\alpha_l^y = \frac{1}{N} \sum_{n=1}^{d} Z_{l\,n}^y, \quad \psi_l^y = \begin{pmatrix} cosh(\alpha_l^y) \\ sinh(\alpha_l^y) \end{pmatrix} \tag{4}$$

where $cosh(\cdot)$ and $sinh(\cdot)$ are the hyperbolic cosine and sine functions respectively. The predictor module is then tasked with directly locating the target in the unit hyperbola given the context embedding and the target patch's positional encoding.

$$\hat{\psi}_l^y = MLP(LayerNorm(z^x + P_l)), \hat{\psi}_l^y \in \mathbb{R}^2 \tag{5}$$

This allows us to frame the learning procedure as a simple regression problem and the whole network can be trained end-to-end (Figure 2f). In practice, we use the smooth L1 loss as the distance function, as it is less sensitive to outliers compared to the L2 loss (Girshick, 2015):

$$L(y, \hat{y}) = \frac{1}{N} \sum_{n=1}^{N} S_n, \quad S_n = \begin{cases} 0.5(y_n - \hat{y}_n)^2/\beta, & \text{if } |y - \hat{y}| < \beta \\ |y - \hat{y}| - 0.5\beta, & \text{otherwise} \end{cases} \tag{6}$$

Thus, we are effectively measuring how far away the context and target patches are in the unit hyperbola of the plane, but the targets are actually described through a high dimensional latent code (Eq. 4). We explicitly show the differences between this choice and using the Euclidean or Hyperbolic distances as energy functions (in the latent space) in Section 4.3. Our proposed pretraining objective forces the context encoder to understand the differences in the hyperbolic angle between the target patches, which can be thought of as establishing a hierarchy.

An asymmetric design of the predictor and target networks (in terms of parameters) is used since it is reported to prevent representation collapse in self-supervised contrastive techniques (Chen et al., 2020; Baevski et al., 2022). We also utilize stop-gradient for the target encoder and update its weights using an Exponential Moving Average (EMA) of the context encoder weights, as done in other works that utilize JEPAs (Assran et al., 2023; Grill et al., 2020; Chen & He, 2021). We now show why these design choices are actually crucial parts of the learning process. Let us make the following assumptions: i) The predictor network is linear; ii) We consider the encoded context features $z^x$ and target coordinates $\psi_l^y$; iii) There is a one-to-one correspondence between context and target patches. (This holds also in practice due to Eq. 5); iv) The problem is treated as a least-squares problem in finite-dimensional vector spaces over the field of reals $\mathbb{R}$. Based on our assumptions, we can rewrite the context features as $X \in \mathbb{R}^{n \times d}$, the target coordinates as $Y \in \mathbb{R}^{n \times 2}$, and the weights of the linear model as $W \in \mathbb{R}^{d \times 2}$. The objective of the predictor is:

$$\underset{W}{\arg\min} \|XW - Y\|^2 \tag{7}$$

where $\|.\|$ indicates the Frobenius norm. The solution to this system can be given by the (multivariate) least squares estimator:

$$W = (X^T X)^{-1} X^T Y \tag{8}$$

By plugging Eq. 8 into Eq. 7 and factorizing $Y$, the least squares solution leads to the error:

$$\left\| (X(X^T X)^{-1} X^T - I_n)Y \right\|^2 \tag{9}$$

Thus, the optimality of the (linear) predictor is defined by the orthogonal projection of $Y$ onto the orthogonal complement of a subspace of $Col(X)$. As is commonly understood, this translates to finding the linear combination of $X$ that is closest, in terms of $\|\cdot\|^2$, to $Y$. Similarly to what was shown in Richemond et al. (2023), we argue that this behavior unveils a key intuition: The target encoder which produces $Y$ must not share weights or be optimized with the same procedure as the context encoder. If that were the case, the easiest solution to Eq. 9 would be using a vector that is orthogonal to itself, i.e., the $\mathbf{0}$ vector, leading to representation collapse. In practice, with a non-linear predictor network, we might immediately run into a degenerate solution where the target encoder and predictor output collapse representations by immediately minimizing the training objective. It is therefore important to parametrize the predictor using a simpler network that is less expressive but can capture the correct dynamics over the training process.

## 4 EXPERIMENTS

The experimental section introduces the empirical evaluation of the Graph-JEPA model in terms of downstream performance on different graph datasets and tasks, as well as various ablation studies that provide more insights into our design choices.

### 4.1 EXPERIMENTAL SETTING

We use the TUD datasets (Morris et al., 2020) as commonly done for graph-level SSL (Suresh et al., 2021; Tan et al., 2023). We utilize seven different graph-classification datasets: PROTEINS, MUTAG, DD, REDDIT-BINARY, REDDIT-MULTI-5K, IMDB-BINARY, and IMDB-MULTI. For all classification experiments, we report the accuracy over five runs (with different seeds) of ten-fold cross validation. It is worth noting that we retrain the Graph-JEPA model for each fold, without ever having access to the testing partition both in the pretraining and fine-tuning stage. For graph regression, we use the ZINC dataset and report the Mean Squared Error (MSE) over ten runs (with different seeds). To produce the unique graph-level representation when fine-tuning, we simply feed all the subgraphs through the trained target encoder and then use mean pooling, obtaining a single feature vector $z_G \in \mathbb{R}^d$ that represents the whole graph. More details regarding the training procedure are available in Appendix A.1.

### 4.2 DOWNSTREAM PERFORMANCE

For the experiments on downstream performance, we follow Suresh et al. (2021) and also report the results of a fully supervised Graph Isomorphism Network (GIN) (Xu et al., 2018), denoted F-GIN. We compare Graph-JEPA against 4 constrastive (Sun et al., 2019; You et al., 2020; Suresh et al., 2021; Hassani & Khasahmadi, 2020), 2 generative (Hou et al., 2022; Tan et al., 2023), and 1 latent (Xie et al., 2022a) graph SSL methods. As can be seen in Table 1, Graph-JEPA achieves competitive results on all datasets, setting the state-of-the-art as a pretrained backbone on five different datasets and coming second on one, out of eight total. Overall, our proposed framework learns semantic embeddings that work well on different types of graphs, showing that Graph-JEPA can indeed be utilized as a general pretraining method for graph-level SSL. Notably, Graph-JEPA works well for both classification and regression and performs better than a supervised GIN on all classification datasets.

We further explore the performance of our model on the synthetic EXP dataset (Abboud et al., 2020). The goal behind this experiment is to empirically verify if Graph-JEPA can learn highly expressive graph representations, in terms of graph isomorphisms. The results in Table 2 show that our model is able to perform much better than commonly used GNNs. Given its local and global exchange of information, this result is to be expected. Most importantly, Graph-JEPA closely rivals the flawless performance of Graph-MLP-Mixer (He et al., 2023), which is trained in a supervised manner.

### 4.3 EXPLORING THE GRAPH-JEPA LATENT SPACE

As discussed in Section 3.4, the choice of energy function has a big impact on the learned representations. Given the latent prediction task of Graph-JEPA, we expect the latent representations to display hyperbolicity. The predictor network is linearly approximating the behavior of the unit hy-

Table 1: Performance of different graph SSL techniques ordered by pretraining type. The results of the competitors are taken as the best values from (Hassani & Khasahmadi, 2020; Suresh et al., 2021; Tan et al., 2023). "-" indicates missing values from the literature. The **best results** are reported in boldface, and the second best are underlined. For the sake of completeness, we also report the results of training GraphMAE on DD, REDDIT-M5, and ZINC in *italics*.

| Model | PROTEINS ↑ | MUTAG ↑ | DD ↑ | REDDIT-B ↑ | REDDIT-M5 ↑ | IMDB-B ↑ | IMDB-M ↑ | ZINC ↓ |
|---|---|---|---|---|---|---|---|---|
| F-GIN | $72.39 \pm 2.76$ | $90.41 \pm 4.61$ | $74.87 \pm 3.56$ | $86.79 \pm 2.04$ | $53.28 \pm 3.17$ | $71.83 \pm 1.93$ | $48.46 \pm 2.31$ | $0.254 \pm 0.005$ |
| InfoGraph | $72.57 \pm 0.65$ | $87.71 \pm 1.77$ | $75.23 \pm 0.39$ | $78.79 \pm 2.14$ | $51.11 \pm 0.55$ | $71.11 \pm 0.88$ | $48.66 \pm 0.67$ | $0.890 \pm 0.017$ |
| GraphCL | $72.86 \pm 1.01$ | $88.29 \pm 1.31$ | $74.70 \pm 0.70$ | $82.63 \pm 0.99$ | $53.05 \pm 0.40$ | $70.80 \pm 0.77$ | $48.49 \pm 0.63$ | $0.627 \pm 0.013$ |
| AD-GCL-FIX | $73.59 \pm 0.65$ | $89.25 \pm 1.45$ | $74.49 \pm 0.52$ | $85.52 \pm 0.79$ | $53.00 \pm 0.82$ | $71.57 \pm 1.01$ | $49.04 \pm 0.53$ | $0.578 \pm 0.012$ |
| AD-GCL-OPT | $73.81 \pm 0.46$ | $89.70 \pm 1.03$ | $75.10 \pm 0.39$ | $85.52 \pm 0.79$ | $54.93 \pm 0.43$ | $72.33 \pm 0.56$ | $49.89 \pm 0.66$ | $\underline{0.544 \pm 0.004}$ |
| MVGRL | - | - | - | $84.5 \pm 0.6$ | - | $74.2 \pm 0.7$ | $51.2 \pm 0.5$ | - |
| GraphMAE | $75.30 \pm 0.39$ | $88.19 \pm 1.26$ | *$74.27 \pm 1.07$* | $88.01 \pm 0.19$ | *$46.06 \pm 3.44$* | $\underline{75.52 \pm 0.66}$ | $\underline{51.63 \pm 0.52}$ | *$0.935 \pm 0.034$* |
| S2GAE | $\mathbf{76.37 \pm 0.43}$ | $88.26 \pm 0.76$ | - | $87.83 \pm 0.27$ | - | $\mathbf{75.76 \pm 0.62}$ | $\mathbf{51.79 \pm 0.36}$ | - |
| LaGraph | $75.2 \pm 0.4$ | $\underline{90.2 \pm 1.1}$ | $\underline{78.1 \pm 0.4}$ | $\underline{90.4 \pm 0.8}$ | $\underline{56.4 \pm 0.4}$ | $73.7 \pm 0.9$ | - | - |
| Graph-JEPA | $\underline{75.67 \pm 3.78}$ | $\mathbf{91.25 \pm 5.75}$ | $\mathbf{78.64 \pm 2.35}$ | $\mathbf{91.99 \pm 1.59}$ | $\mathbf{56.73 \pm 1.96}$ | $73.68 \pm 3.24$ | $50.69 \pm 2.91$ | $\mathbf{0.434 \pm 0.014}$ |

Table 2: Classification accuracy on the synthetic EXP dataset (Abboud et al., 2020), which contains 600 pairs of non-isomorphic graphs that are indistinguishable by the 1-WL test. The competitor models (Kipf & Welling, 2016a; Bresson & Laurent, 2017; Xu et al., 2018; Dwivedi & Bresson, 2020) and their results are from the work of He et al. (2023).

| Model | GCN | GatedGCN | GINE | GraphTransformer | Graph-MLP-Mixer | Graph-JEPA |
|---|---|---|---|---|---|---|
| Accuracy | $51.90 \pm 1.96$ | $51.73 \pm 1.65$ | $50.69 \pm 1.39$ | $52.35 \pm 2.32$ | $\mathbf{100.00 \pm 0.00}$ | $\underline{98.77 \pm 0.99}$ |

perbola such that it best matches with the generated target coordinates (Eq. 6). Thus, the network is actually trying to estimate a space that can be considered a particular section of the hyperboloid model (Reynolds, 1993), where hyperbolas appear as geodesics. We are therefore evaluating our energy in a (very) restricted part of hyperbolic space. As mentioned before, we find this task to offer great flexibility as it is straightforward to implement and it is computationally efficient compared to the hyperbolic distance used to typically learn hyperbolic embeddings in the Poincaré ball model (Nickel & Kiela, 2017).

Table 3 provides empirical evidence for our conjectures regarding the suboptimality of Euclidean or Poincaré embeddings. The results reveal that learning the distance between patches in the 2D unit hyperbola provides a simple way to get the advantages of both embedding types. Hyperbolic embeddings must be learned in lower dimensions due to stability issues (Yu & De Sa, 2021), while Euclidean ones do not properly reflect the dependencies between subgraphs and the hierarchical nature of graph-level concepts. Our results suggest that the hyperbolic (Poincaré) distance seems to generally be a better choice than the Euclidean distance in lower dimensions, but it is very unstable and expensive computationally in high dimensions. The proposed approach provides the best overall results, with the Euclidean distance performing negligibly better in a single case. We provide a qualitative example that reveals how the embedding space is altered when using our proposed latent objective in Appendix A.3.

## 4.4 ABLATION STUDIES

In the following, we present various ablation studies to get a better understanding of the different design choices of Graph-JEPA. For all ablations, we consider 4 out of the 8 datasets initially pre-

Table 3: Comparison of Graph-JEPA performance for different distance functions. The optimization for Poincaré embeddings in higher dimensions is problematic, as shown by the NaN loss on the IMDB-B dataset. LD stands for Lower Dimension, where we use a smaller embedding size than in the previous cases. As a reminder to the reader, the results on ZINC report the MSE.

| Distance function | Ours | Euclidean | Hyperbolic | Euclidean (LD) | Hyperbolic (LD) |
|---|---|---|---|---|---|
| MUTAG | $\mathbf{91.25 \pm 5.75}$ | $87.04 \pm 6.01$ | $89.43 \pm 5.67$ | $86.63 \pm 5.9$ | $86.32 \pm 5.52$ |
| REDDIT-M | $\mathbf{56.73 \pm 1.96}$ | $56.55 \pm 1.94$ | $56.19 \pm 1.95$ | $54.84 \pm 1.6$ | $55.07 \pm 1.83$ |
| IMDB-B | $73.68 \pm 3.24$ | $\mathbf{73.76 \pm 3.46}$ | NaN | $72.5 \pm 3.97$ | $73.4 \pm 4.07$ |
| ZINC | $\mathbf{0.434 \pm 0.01}$ | $0.471 \pm 0.01$ | $0.605 \pm 0.01$ | $0.952 \pm 0.05$ | $0.912 \pm 0.04$ |

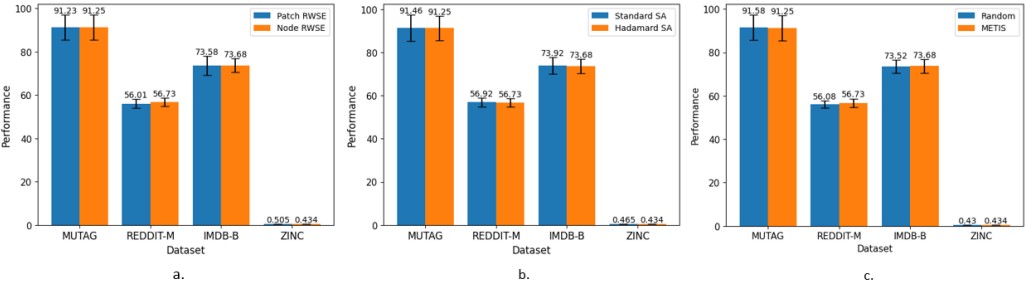

Figure 3: Ablation studies: a. Performance when using absolute (node-level) vs relative (patch-level) RWSEs. b. Results when using the standard self-attention mechanism vs a graph-specific one. c. Performance when using METIS subgraphs vs using random subgraphs. We remind the reader that the MAE is reported on ZINC and Accuracy on the other datasets. The error bars represent the average standard deviation values over the runs in order to report results similarly to the tables in previous sections. Best viewed in color.

sented in Table 1, making sure to choose different graph types for a fair comparison. All results are summarized in Figure 3, but we also provide the results in tabular format in Appendix A.2 for clarity.

**Positional embedding.** Following He et al. (2023), it is possible to use the RWSE of the patches as conditioning information. Formally, let $B \in \{0, 1\}^{p \times N}$ be the patch assignment matrix, such that $B_{ij} = 1$ if $v_j \in p_i$. We can calculate a coarse patch adjacency matrix $A' = BB^T \in \mathbb{R}^{p \times p}$, where each $A'_{ij}$ contains the node overlap between $p_i$ and $p_j$. The RWSE can be calculated as described in Eq. 1 (considering $A'$). We test Graph-JEPA with these relative positional embeddings and find that they still provide good performance, but consistently fall behind the node-level (global) RWSE that we employ in our formulation (Figure 3a). An issue of the relative embeddings is that the number of shared neighbors obscures the local peculiarities of each patch, adding more variance to the results.

**Using different Self-Attention mechanisms.** As stated in Section 3.3, Graph-JEPA uses Hadamard self-attention (He et al., 2023), which provides a strong inductive bias for graphs. It is possible to make no assumptions about the data and render the prediction task more challenging, by using the standard self-attention mechanism (Vaswani et al., 2017). We show the results of this change in Figure 3b. The results reveal that the performance is slightly better but more unstable, i.e., additional variance in the results. This is to be expected since, as mentioned previously, using the standard self-attention mechanism emphasizes a lack of inductive bias regarding the input data.

**Random subgraphs.** A natural question that arises in our framework is how to design the spatial partitioning procedure. Using a structured approach like METIS is intuitive and leads to favorable results. Another option would be to simply extract random, non-empty subgraphs as context and target. As can be seen in Figure 3c, the random patches provide strong performance as well but do not show any improvement. Even though our results show that it might not be necessary to use a structured way to extract the patches, not all graphs are equally easy to sample randomly from. Thus, we advocate extracting subgraphs with METIS as it is a safer option in terms of generalizability across different graphs and what inductive biases it provides.

## 5 CONCLUSION

In this work, we introduce the first Joint Embedding Predictive Architecture (JEPA) for graph-level Self-Supervised Learning (SSL). A proper design of the model both in terms of data preparation and pretraining objective reveals that it is possible for a neural network model to self-organize the semantic knowledge embedded in a graph, demonstrating competitive performance in both graph classification and regression. Future research directions include extending the method to node and edge-level learning, exploring the expressiveness of Graph-JEPA, and gaining insights into the optimal geometry of the embedding space for graph SSL.

## 6 Ethics and Reproducibility Statement

We acknowledge the significance of ethical considerations and reproducibility in our research, and in this section, we address these aspects explicitly.

Regarding ethics, Graph-JEPA serves as a general pre-training model and has undergone validation through a benchmarking approach. Our methodology exclusively employs publicly available datasets and libraries, with no utilization of private information. Additionally, we have prioritized the development of an efficient model that requires minimal computational resources. This approach aligns with the increasing awareness of the environmental impact associated with deep learning models.

Concerning reproducibility, we have taken deliberate steps to ensure clarity and motivation for our general architecture and design choices throughout the paper. Appendix A.1 provides comprehensive details about the tools used in implementing Graph-JEPA, the experimental setup, and hyperparameters. Furthermore, we offer a visual representation of the model's objective in Appendix A.3. Lastly, we commit to making the Graph-JEPA code publicly available upon acceptance, facilitating its use within the broader research community.

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

# A  APPENDIX

## A.1  EXPERIMENTAL SETTING

For the datasets that do not natively have node and edge features, we use a simple constant (0) initialization. The subgraph embedding GNN (Figure 2c) consists of the GIN operator with support for edge features (Hu et al., 2019) a.k.a GINE. For fine-tuning, we feed the high-dimensional representation of the hyperbolic angle produced by the target encoder to a linear model with L2 regularization. Specifically, we employ Logistic Regression with L2 regularization on the classification datasets, and for the ZINC dataset, we utilize Ridge Regression. We also provide details regarding the hyperparameters of the JEPA design for Graph-JEPA in Table 4. The neural network is trained using the Adam optimizer (Kingma & Ba, 2014). The neural network modules were implemented using PyTorch (Paszke et al., 2019) and PyTorch-Geometric (Fey & Lenssen, 2019), while the linear classifiers and cross-validation procedure were implemented with the Scikit-Learn library (Pedregosa et al., 2011). All experiments were performed on a single Nvidia RTX 3090 GPU.

Table 4: Hyperparameters regarding the Graph-JEPA design for the TUD datasets.

| Hyperparameter | PROTEINS | MUTAG | DD | REDDIT-B | REDDIT-M5 | IMDB-B | IMDB-M | ZINC |
|---|---|---|---|---|---|---|---|---|
| # Subgraphs | 32 | 32 | 32 | 128 | 128 | 32 | 32 | 32 |
| # GNN Layers | 2 | 2 | 3 | 2 | 2 | 2 | 2 | 2 |
| # Encoder Blocks | 4 | 4 | 4 | 4 | 4 | 4 | 4 | 4 |
| Embedding size | 512 | 512 | 512 | 512 | 512 | 512 | 512 | 512 |
| RWSE size | 20 | 15 | 30 | 40 | 40 | 15 | 15 | 20 |
| # context - # target | 1 - 2 | 1 - 3 | 1 - 4 | 1 - 4 | 1 - 4 | 1- 4 | 1- 4 | 1- 4 |

## A.2  ABLATION RESULTS

Tables 5, 6, and 7 present the results of all ablation studies in tabular format for additional clarity. We also report some additional experiments that reflect the benefits of Graph-JEPA. In Table 8, we compare the total training time of the SSL task in order to provide the representations that prove optimal for the downstream task. Table 9 on the other hand contains the results of parametrizing the whole architecture, other than the initial GNN encoder, through MLPs. Graph-JEPA manages to perform well even in this scenario, albeit the proposed version with the Transformer encoders performs better.

Table 5: Performance when using absolute (node-level) vs relative (patch-level) RWSEs.

| Dataset | Node-level RWSE | Patch-level RWSE |
|---|---|---|
| MUTAG | **91.25 ± 5.75** | 91.23 ± 5.86 |
| REDDIT-M | **56.73 ± 1.96** | 56.01 ± 2.1 |
| IMDB-B | **73.68 ± 3.24** | 73.58 ± 4.47 |
| ZINC | **0.434 ± 0.01** | 0.505 ± 0.005 |

Table 6: Performance when extracting subgraphs with METIS vs. using random subgraphs.

| Dataset | METIS | Random |
|---|---|---|
| MUTAG | $91.25 \pm 5.75$ | $\mathbf{91.58 \pm 5.82}$ |
| REDDIT-M | $\mathbf{56.73 \pm 1.96}$ | $56.08 \pm 1.69$ |
| IMDB-B | $\mathbf{73.68 \pm 3.24}$ | $73.52 \pm 3.08$ |
| ZINC | $0.434 \pm 0.01$ | $\mathbf{0.43 \pm 0.01}$ |

Table 7: Performance when using the standard self-attention mechanism vs. a graph-specific one.

| Dataset | Standard Attention | Hadamard Attention |
|---|---|---|
| MUTAG | $\mathbf{91.46 \pm 6.1}$ | $91.25 \pm 5.75$ |
| REDDIT-M | $\mathbf{56.92 \pm 2.09}$ | $56.73 \pm 1.96$ |
| IMDB-B | $\mathbf{73.92 \pm 3.85}$ | $73.68 \pm 3.24$ |
| ZINC | $0.465 \pm 0.01$ | $\mathbf{0.434 \pm 0.01}$ |

Table 8: Total training time of MVGRL, GraphMAE, and Graph-JEPA for pretraining (single run) based on the optimal configuration for downstream performance. OOM stands for Out-Of-Memory.

| Dataset | Model | Training time |
|---|---|---|
| IMDB | MVGRL | $\sim 7$ min |
| | GraphMAE | $\sim 1.5$ min (1min 36sec) |
| | Graph-JEPA | $< \mathbf{1min}$ **(56 sec)** |
| REDDIT-M5 | MVGRL | *OOM* |
| | GraphMAE | $\sim 46$ min |
| | Graph-JEPA | $\sim \mathbf{18}$ **min** |

Table 9: Performance when parametrizing the context and target encoders through MLPs vs using the proposed Transformer encoders.

| Dataset | Transformer Encoders | MLP Encoders |
|---|---|---|
| MUTAG | $91.25 \pm 5.75$ | $90.82 \pm 6.1$ |
| REDDIT-B | $91.99 \pm 1.59$ | $89.79 \pm 2.37$ |
| ZINC | $0.465 \pm 0.01$ | $0.47 \pm 0.01$ |

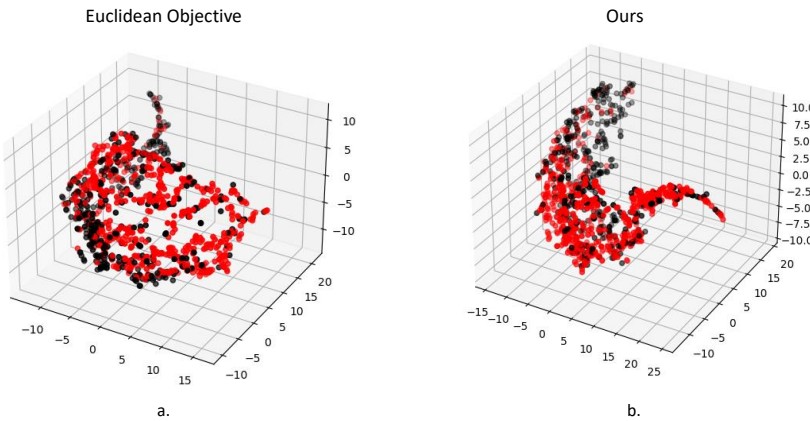

Figure 4: 3D representation (via t-SNE) of the latent codes used to finetune the linear classifier for the DD dataset. Best viewed in color.

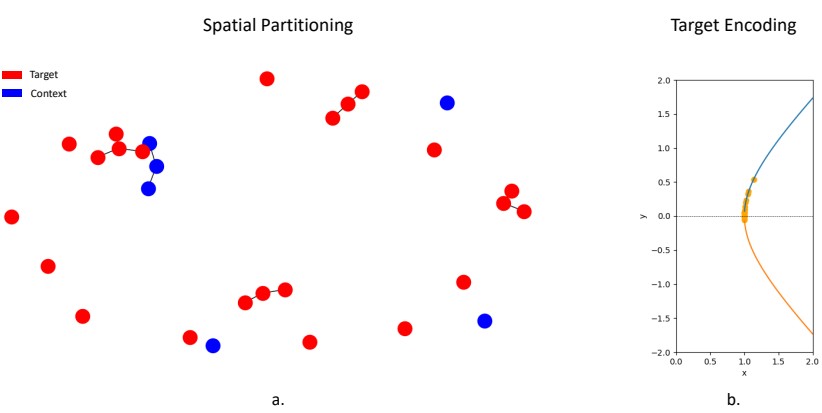

Figure 5: Visualization of a partitioned graph from the ZINC dataset and its corresponding embedding in the 2D unit hyperbola, as detailed in Eq. 4

## A.3 VISUALIZATIONS

In this section, we provide two qualitative examples that give a more visual understanding of how and what Graph-JEPA learns. Firstly, we plot a low-dimensional representation of the high-dimensional latent code obtained from the target encoder when using the Euclidean objective vs using our objective in Figure 4. The modified prediction task greatly alters the geometry and distribution of the latent codes, avoiding the formation of a blob-like structure like in the Euclidean distance task.

Figure 5 depicts how the Graph-JEPA prediction task is framed. In 5a we can see a particular molecule from the ZINC dataset and the corresponding context and target nodes. The target encoder learns a high-dimensional embedding which then can represent the nodes in the 2D unit hyperbola, as shown in 5b. The predictor network tries to guess these coordinates and this training procedure gives rise to the representations learned in Figure 4b.

