# OpenReview forum: "Graph-level Representation Learning with Joint-Embedding Predictive Architectures"
_ICLR.cc/2024/Conference — Submitted to ICLR 2024_

### Official Review · Reviewer_dWk6 · 2023-10-25

**Soundness:** 3 good
**Presentation:** 2 fair
**Contribution:** 3 good
**Rating:** 6
**Confidence:** 4

**Summary:**

This work proposes a new self-supervised technique for graph neural networks. Grounded on joint-embedding predictive architecture (JEPAs), the proposed Graph-JEPA is designed to predict the latent embeddings for multiple subgraphs based on a random subgraph. Experiments are performed on graph-level tasks.

**Strengths:**

1.	The design of the loss objective is good.

2.	The ablation studies and discussion are detailed and insightful.

**Weaknesses:**

1.	Missing literature in related work. In addition to contrastive methods and generative methods, self-supervised graph representation should also include existing predictive methods [1]. For example, CCA-SSG [2] and LaGraph [3] are two existing works using latent embedding prediction. Such predictive methods should be discussed in related works, and they should be used as baseline methods to compare results.

2.	The performance improvement is marginal based on the main results in Table 1.

3.	Most existing SSL methods can handle both graph-level and node-level tasks. However, the proposed Graph-JEPA only supports graph-level downstream tasks.

[1]. Xie, Yaochen, et al. "Self-supervised learning of graph neural networks: A unified review." IEEE transactions on pattern analysis and machine intelligence 45.2 (2022): 2412-2429.

[2]. Zhang, Hengrui, et al. "From canonical correlation analysis to self-supervised graph neural networks." Advances in Neural Information Processing Systems 34 (2021): 76-89.

[3]. Xie, Yaochen, Zhao Xu, and Shuiwang Ji. "Self-supervised representation learning via latent graph prediction." International Conference on Machine Learning. PMLR, 2022.

**Questions:**

1.	Authors claim that the Graph-JEPA is more efficient than contrastive methods since it doesn’t require data augmentations or negative samples. I’m wondering how efficient it is. Could you add a quantitative comparison for the efficiency?

2.	The proposed Graph-JEPA uses Transformer encoder blocks. However, most baseline models are based on simpler models like GIN and GCN. Is it an unfair comparison? Can you use GIN/GCN encoder?

---

> ### Author Response · Authors · 2023-11-15
> **Reponse to Reviewer dWk6**
>
> We would like to sincerely thank the reviewer for the detailed comments and valuable questions, and also for positively evaluating our loss function design and analysis. We provide details to clarify the major concerns.
>
> > Q1: Missing literature in related work
>
> A1: We thank the reviewer for this attentive comment, we have indeed missed these important works talking about latent graph prediction. All three suggested papers have been added in the Related Works section (Section 2) of the rebuttal revision, along with a small discussion. As it is the most recent and competitive work, we have added results from LaGraph in Table 1 of the rebuttal revision as well.
>
> > Q2:  The performance improvement is marginal based on the main results in Table 1. Could you add a quantitative comparison for the efficiency?
>
> A2: We provide a discussion regarding our performance in the response to Q2 of reviewer 15Jn.  We provide the same explanation here in order for the reviewer to have a self-contained response. Given that another big advantage of Graph-JEPA is efficiency, and the reviewer had asked about experiments regarding this, we provide in this answer the *results of the comparison for the efficiency*: Graph-JEPA provides a new way of thinking about SSL on graphs, by essentially bridging the advantages of both contrastive and generative approaches. Contrastive approaches present several computational bottlenecks, while generative models such as GraphMAE tend to strongly overfit because of the generative objective. As mentioned in our paper, we can bypass the need for data augmentation and negative samples, while not needing to fit the data distribution. Not only does this bring about a new paradigm and research direction for graph SSL, but we also show that it can be beneficial in terms of performance and expressiveness of representations. Additionally, we provide new results in this response and in the appendix of the rebuttal revision showcasing the efficiency of Graph-JEPA compared to GraphMAE and MVGRL. The table below presents the training time needed for the three methods in order to produce the representations that give rise to optimal downstream performance. Our model clearly shows superior efficiency and convergence, especially as the size of the dataset grows ( REDDIT-MULTI-5K).
>
> | Dataset   | Model       | Training time          |
> |-----------|-------------|------------------------|
> | IMDB      | MVGRL       | ∼ 7 min                |
> |           | GraphMAE    | ∼ 1.5 min (1min 36sec) |
> |           | Graph-JEPA  | **< 1min (56 sec)**        |
> | REDDIT-MULTI-5K | MVGRL       | OOM                    |
> |           | GraphMAE    | ∼ 46min                |
> |           | Graph-JEPA  | **∼ 18min**                |
>
>
> > Q3:  Most existing SSL methods can handle both graph-level and node-level tasks. However, the proposed Graph-JEPA only supports graph-level downstream tasks.
>
> A3: While we agree with the statement of the reviewer, our method is framed with the graph-level representation learning problem in mind, therefore performance on node-level tasks was not taken into account. Graph-level learning is crucial in different domains and most importantly, has different dynamics, particularly in terms of hierarchy, compared to node-level concepts. Our self-predictive task and objective function are both implemented considering these aspects for an improved graph-level representation. Nevertheless, this is definitely a direction of future work for future applications of JEPAs in the graph domain, for which we hope our work can provide inspiration.
>
> > Q4: The proposed Graph-JEPA uses Transformer encoder blocks. However, most baseline models are based on simpler models like GIN and GCN. Is it an unfair comparison? Can you use GIN/GCN encoder?
>
> A4: We utilize the Transformer to exchange information between the subgraph embeddings because of the modeling flexibility, without necessarily requiring an adjacency matrix between the various subgraphs. Firstly, we argue that the comparison is not unfair since we utilize a MP-GNN to encode the node features before moving on to the subgraph encoding. To demonstrate the benefits of our architecture and SSL objective, we provide the downstream results of a Graph-JEPA that only utilizes MLP modules after the MP-GNN subgraph encoding. The results in the Table below, show that the model remains competitive, maintaining state-of-the-art performance on MUTAG, REDDIT-BINARY, and ZINC, albeit marginally in the first two cases. Nevertheless, even when parametrized through simple MLPs, Graph-JEPA is still able to outperform most contrastive approaches presented in Table 1 of the submission.
>
> | Dataset  | Transformer Encoders | MLP Encoders     |
> |----------|----------------------|------------------|
> | MUTAG    | 91.25 $\pm$ 5.75     | 90.82 $\pm$ 6.10  |
> | REDDIT-B | 91.99 $\pm$ 1.59     | 89.79 $\pm$ 2.37 |
> | ZINC     | 0.465 $\pm$ 0.01     | 0.47 $\pm$ 0.01 |

---

> > ### Comment · Reviewer_dWk6 · 2023-11-22
> >
> > Thanks for the response and additional results. I've raised the score to weak accept.

---

### Official Review · Reviewer_15Jn · 2023-10-28

**Soundness:** 2 fair
**Presentation:** 2 fair
**Contribution:** 2 fair
**Rating:** 3
**Confidence:** 4

**Summary:**

The authors propose Graph-JEPA. Graph-JEPA uses two encoders to receive the input and one of the encoders predicts the latent representation of the input signal based on another encoder.

**Strengths:**

The writing is clear. First JEPA for graph. The authors provide an analysis to explain why JEPA works for the graph domain.

**Weaknesses:**

1. The method is not novel. The proposed Graph-JEPA is very similar to MLM in BERT, which utilizes the context to predict the masked word type.

2.  The proposed method is too simple and the motivation is not clear. We have graph MAE and contrastive learning. Why do we need JEPA for the graph domain?

3. Compared with graph MAE and S2GAE, the performance is not good enough to show it can inspire future research.

**Questions:**

N/A

---

> ### Author Response · Authors · 2023-11-15
> **Reponse to Reviewer 15Jn**
>
> We would like to express our gratitude to the reviewer for acknowledging our writing and analysis. We proceed by answering the listed weaknesses of our paper.
>
> > Q1: The method is not novel. The proposed Graph-JEPA is very similar to MLM in BERT, which utilizes the context to predict the masked word type.
>
> A1: While the idea of masking a part of the input can be considered similar to BERT, Graph-JEPA is quite different as a pretraining strategy. MLM with BERT *is an autoencoding procedure*, thus it is actually part of the generative pretraining family of SSL, like most Language Modeling approaches. Our approach works in *latent space only* and it’s predictive of the latent embeddings, thereby it does not fit the data distribution. Naturally, this completely alters learning dynamics. A general overview, both schematic and written was made available in the Related Work (Section 2) of our submission.
>
> > Q2: The proposed method is too simple and the motivation is not clear. We have graph MAE and contrastive learning. Why do we need JEPA for the graph domain?
>
> A2: Graph-JEPA provides a new way of thinking about SSL on graphs, by essentially bridging the advantages of both contrastive and generative approaches. Contrastive approaches present several computational bottlenecks, while generative models such as GraphMAE tend to strongly overfit because of the generative objective. As mentioned in our paper, we can bypass the need for data augmentation and negative samples, while not needing to fit the data distribution. Not only does this bring about a new paradigm and research direction for graph SSL, but we also show that it can be beneficial in terms of performance and expressiveness of representations. Additionally, we provide new results in this response and in the appendix of the rebuttal revision showcasing the efficiency of Graph-JEPA compared to GraphMAE and MVGRL. The table below presents the training time needed for the three methods in order to produce the representations that give rise to optimal downstream performance. Our model clearly shows superior efficiency and convergence, especially as the size of the dataset grows (REDDIT-MULTI-5K).
>
> | Dataset   | Model       | Training time          |
> |-----------|-------------|------------------------|
> | IMDB      | MVGRL       | ∼ 7 min                |
> |           | GraphMAE    | ∼ 1.5 min (1min 36sec) |
> |           | Graph-JEPA  | **< 1min (56 sec)**        |
> | REDDIT-MULTI-5K | MVGRL       | OOM                    |
> |           | GraphMAE    | ∼ 46min                |
> |           | Graph-JEPA  | **∼ 18min**                |
>
>
> > Q3: Compared with graph MAE and S2GAE, the performance is not good enough to show it can inspire future research.
>
> A3: Graph-JEPA outperforms Graph-MAE and S2GAE by a significant margin on MUTAG, DD, and REDDIT-BINARY. We have added the results of using GraphMAE with the officially provided code on REDDIT-MULTI-5K and ZINC in the revised version of the paper, where Graph-JEPA still shows superior performance. Our model remains competitive in all datasets, setting the state of the art in 5 datasets and coming second in 1, out of 8 total datasets. We obtain these results while also showing great efficiency, as demonstrated in the table above. We believe all of these factors make Graph-JEPA a valid candidate for the exciting new frontier of JEPAs in graph SSL.

---

> ### Author Response · Authors · 2023-11-22
> **A gentle reminder for reviewer 15Jn**
>
> Dear reviewer 15Jn,
>
> As the discussion deadline approaches, we kindly seek clarification on whether the three weaknesses highlighted in the initial review have been adequately addressed in our rebuttal. We have endeavored to furnish comprehensive details in each response to facilitate your assessment. If further clarification is needed or if there are additional concerns regarding our paper, we would greatly appreciate the opportunity to engage in further discussion. Thank you once again for your valuable comments and feedback.

---

### Official Review · Reviewer_b3AV · 2023-11-01

**Soundness:** 3 good
**Presentation:** 1 poor
**Contribution:** 1 poor
**Rating:** 3
**Confidence:** 4

**Summary:**

This paper proposes Graph-JEPA, the first Joint-Embedding Predictive Architectures (JEPAs) for the graph domain.
The application of JEPA to graphs seems to be novel.

**Strengths:**

- The proposed method is technically sound
- Based on the experimental results, the improvement between Graph-JEPA over the baselines seems to be strong

**Weaknesses:**

- The overall method seems to be a direct application of JEPA to graphs.
- The discussion of "why does graph-JPEA works" is useful, but not information. Any theoretical analysis here will be useful.
- The experimental settings are confusing. It is unclear to me why "GCN", a GNN model, can be compared with "Graph-JEPA", which is a graph self-supervised training method.
- All the figures and tables are not professional and could be improved to be more appealing. Font sizes and colors should be improved.

**Questions:**

- What makes applying JEPA to graphs special and non-trivial?

---

> ### Author Response · Authors · 2023-11-15
> **Reponse to Reviewer b3AV**
>
> We would like to thank the reviewer for their positive comments regarding the performance and soundness of our method. In the following, we aim to provide detailed responses to the raised weaknesses and questions.
>
> > Q1: The overall method seems to be a direct application of JEPA to graphs. What makes applying JEPA to graphs special and non-trivial?
>
> A1: The Joint-Embedding Predictive Architecture is a recently proposed, general design for SSL [1]. As such, applying this concept to the graph domain brings about several challenges. This is very similar to previous challenges faced when initially applying contrastive learning or autoencoding in the graph domain, tackled by related work. We state the main challenges that Graph-JEPA tries to solve below:
> 1. How do we design the self-supervised task in order for it to be self-predictive and expressive? This includes creating supervision from the input graphs such that an appropriate design can capture feature similarities via latent reconstruction. There are multiple possibilities: Is it appropriate to work only at node level, or at structure level? If we decide to operate at subgraph level, how do we extract subgraphs? Should there be an overlap between them? How should the SSL pretraining task be concretely defined? Given all these possibilities, developing an intuitive and simple to implement JEPA is not straightforward.
>  2. How can we make sure that the SSL learning procedure defined in step 1 is optimal for graph-level concepts? First of all, we want our representations to not collapse in the latent space, and we also wish to impose a particular geometry to the latent space that reflects the hierarchical nature of graph-level concepts. Both of these problems require an appropriately designed solution
> 3. Can we learn expressive graph representations through a JEPA, such that we bypass the 1-WL expressiveness limit of common Message Passing Graph Neural Networks (MP-GNNs)? Given that we wish to lean graph-level representations, it is imperative that these are expressive in terms of graph isomorphisms and can go beyond simple MP-GNNs.
>
> Our proposed architecture tackles all these questions and reveals that it is possible to build a JEPA architecture for graph-level SSL, as shown by the competitive performance both in downstream classification and expressiveness.
>
> > Q2: The discussion of "why does graph-JPEA works" is useful, but not information. Any theoretical analysis here will be useful.
>
> A2: The title of the section might have been misleading so we have removed it in the rebuttal revision. The content has been merged in two different parts: the additional theoretical analysis on the necessity of the stop-gradient operation and the asymmetric predictor network has been added in the end of Section 3.4, while the hyperbolic properties of the latent space are discussed in Section 4.3. Our aim with these discussions is to show that the design choices behind our proposed approach make sense theoretically and not just empirically. We agree with the reviewer that an additional theoretical analysis as to why a JEPA architecture can learn in the first place would be useful, but that is beyond the scope of this work. We believe that the insights from our paper provide information regarding key design aspects of the architecture and we would happily discuss further the conclusions drawn in the paper if the reviewer would kindly provide more details regarding what aspects they find not informative.
>
> > Q3: The experimental settings are confusing. It is unclear to me why "GCN", a GNN model, can be compared with "Graph-JEPA", which is a graph self-supervised training method.
>
> A3: We apologize for the lack of clarity or confusion caused by the experimental setting. GCN only appears in Table 2, where we empirically evaluate the expressiveness of the learned representations of the different models. Given that most MP-GNNs are known to be as expressive as the 1-WL test, the experiments indeed show that they are incapable of recognizing 1-WL indistinguishable graphs, with an accuracy level (approximately 50%) that shows almost a tendency to randomly guess. In contrast, representations learned by Graph-JEPA are able to perform almost perfectly. Thus the use of GCN, similar to the use of GIN in Table 1 of the submission, is necessary simply to put the results into perspective. The same study is performed in [2].
>
> > Q4: All the figures and tables are not professional and could be improved to be more appealing. Font sizes and colors should be improved.
>
> A4: We are sorry for the seemingly unprofessional presentation of our work. We have updated all of the tables in the rebuttal revision and tried to provide a better layout.
>
> [1] LeCun, Yann. "A path towards autonomous machine intelligence version 0.9. 2, 2022-06-27." Open Review 62 (2022).
>
> [2] He, Xiaoxin, et al. "A generalization of vit/mlp-mixer to graphs." International Conference on Machine Learning. PMLR, 2023.

---

> ### Author Response · Authors · 2023-11-22
> **A gentle reminder for reviewer b3AV**
>
> Dear reviewer b3AV,
>
> As the discussion deadline approaches, we kindly seek clarification on whether we have adequately answered your question and addressed the four weaknesses highlighted in the initial review. We have made diligent efforts, including modifications to the paper, adjustments to the paper's formatting, and answering every single weak point. If additional clarification is required or if there are further concerns with our paper, we would greatly appreciate the opportunity for additional discussion.
>
> Thank you for your constructive feedback.

---

### Author Response · Authors · 2023-11-15
**Summary of the revised version**

We sincerely thank all the reviewers for their assessments of our paper. We have revised the submission based on the provided reviews. Below is the list of the main changes:

1. We have removed the "Why does Graph-JEPA work" subsection (previously Section 3.5) and have separated the contents present in two different sections, in order to avoid any confusion given by the title. The theoretical analysis of the necessity of the stop-gradient operation and the asymmetric predictor network has been added at the end of Section 3.4, while the hyperbolic properties of the latent space are discussed in Section 4.3.
2. We have added the following results: Table 1 has been updated by adding the performance obtained by GraphMAE on the datasets where the original paper does not provide results (DD, REDDIT-MULTI-5K, and ZINC). We used the [original code](https://github.com/THUDM/GraphMAE/tree/pyg) and the hyperparameters therein for these results. Additionally, the performance of LaGraph [1] has also been reported in Table 1.  Finally, we have added additional numerical comparisons of training efficiency and results when using MLP context and target encoders in the Appendix.
3. We have added the three works mentioned by Reviewer dWk6 in the Related Work section (Section 2) along with a short discussion regarding the similarities they share with Graph-JEPA.
4. The formatting of all the tables has been changed so that it is more minimal, and hopefully more aesthetically pleasing.

[1]. Xie, Yaochen, Zhao Xu, and Shuiwang Ji. "Self-supervised representation learning via latent graph prediction." International Conference on Machine Learning. PMLR, 2022.

---

### Author Response · Authors · 2023-11-21
**A gentle reminder**

Dear Reviewers,

Thank you for your time and effort in reviewing our paper. We have tried to answer the constructive questions and valuable feedback we got, however, we have not received any responses. We would be extremely grateful if you could take some time to go through our answers and the revised version of our paper, as the discussion stage only has one day left.

We are looking forward to any further comments on our work.

Best,
Authors

---

### Meta-Review · Area_Chair_PNZY · 2023-12-06

**Metareview:**

This work proposes the Graph-JEPA method, which is presented as the first application of Joint-Embedding Predictive Architectures (JEPA) to the graph domain. The pitch of the work is interesting and the results potentially compelling. However, several issues were raised by the reviewers, which the Authors attempted to address in a rebuttal which includes additional baselines, contextualises within further related work, and generally improves the clarity.

I need to start by stating my disappointment with most of the Reviewers' responses in this work. In spite of multiple repeated pings, both by myself and by the Authors, only one Reviewer responded to the Authors' rebuttal, and did so in a somewhat uninformative way. In absence of a fruitful discussion, I had no choice but to make an independent judgement call, based on my own reading of the paper, the points made by the reviewers, and the responses by the Authors.

Unfortunately, having read the paper myself (along with a few of its citations) and contextualised its contribution against what the Reviewers claim, I need to concur with the majority-vote, and argue for rejection for this work.

As presented, this work follows a 'tried-and-tested' template of applying a method which has seen success in self-supervised learning on images and deploys it in the graph domain, in a more-or-less direct way. This spans works like DGI (based on DIM), GRACE (based on SimCLR), BGRL (based on BYOL), Graph-BT (based on Barlow Twins)... etc. The clarity of such a pipeline implies that, to qualify as a worthwhile contribution at a top-tier venue like ICLR, the method should go beyond exposing the application, and make conclusive evidence that there are unique benefits to this particular variation of SSL dataflow.

In this regard, while there are some noteworthy results in the paper, I believe the Authors need to be more careful with the contextualisation of their claims.

First: the authors claim their method is _"the first JEPA for the graph domain"_, but also write _"The most well-known works to use particular instantiations of JEPAs before the term was officially coined are BYOL (Grill et al., 2020)"_. These two sentences are contradictory, as evidenced by BGRL (Thakoor et al., ICLR'22), which directly adapts BYOL to the graph domain; however, methods like BGRL are not cited in the present work. While this issue was not explicitly raised by reviewers, I don't think a method like BGRL would have been particularly challenging to find in literature review -- a simple Google search of _"BYOL Graph"_ provides BGRL as the first two results.

The absence of BGRL would not, on its own, be grounds to recommend rejection, as it could potentially be explained by the fact that BGRL mainly evaluates on node-level tasks but Graph-JEPA is pitched as mainly a graph-level method. One of the Reviewers explicitly pointed out the somewhat puzzling focus on graph-level tasks only, to which the Authors replied:

> Graph-level learning is crucial in different domains and most importantly, has different dynamics, particularly in terms of hierarchy, compared to node-level concepts. Our self-predictive task and objective function are both implemented considering these aspects for an improved graph-level representation.

However, while this could be a plausible response, I don't think that this absolves the authors from comparing against a relevant node-level method. Any node-level graph SSL method can easily be piped into a graph classifier -- simply aggregate the learnt node embeddings and classify the aggregated vector. I would have hoped to see at least one such comparison, to actually confirm that all the effort that went into designing the custom objective function actually translated into concrete gains.

The added comparison with LaGraph could constitute such a comparison, as LaGraph is in many ways similar to BGRL. However, due to the massive error bars / instability of the proposed method, _there is not a single dataset where I see a statistically-significant outperformance of JEPA vs. LaGraph_. Hence, I do not see conclusive evidence for a JEPA-style handcrafted graph-level loss as opposed to adapting a (combination of) node-level ones.

All of the above leave me with no conclusive evidence for why Graph-JEPA should be used instead of related methods, as it's looking like we specialise further the JEPA template (which appears not to have been ported to graphs for the first time, following the chain-of-thought given by the Authors' writing), with no obviously significant returns compared to LaGraph/BGRL-style methods, and no clear theoretical evidence for why these losses are more favourable than the already existing solutions.

Let me conclude this review by saying that there is clear potential to the Authors' work, and I do hope it will get published at a top-tier venue. However, as it stands, I unfortunately cannot recommend acceptance.

**Justification For Why Not Higher Score:**

Unsubstantiated claims about this being the first JEPA-style work in the graph domain, lack of conclusive evidence that a specialised graph-level objective is necessary (high variance in the predictions), and no direct comparative theoretical evidence to favour this method over competing ones (BGRL, LaGraph, etc.).

**Justification For Why Not Lower Score:**

N/A

---

### Decision · Program_Chairs · 2024-01-16

Reject